# Mechanistic insights into volatile anesthetic modulation of K2P channels

Aboubacar Wague[1], Thomas T Joseph[2], Kellie A Woll[2], Weiming Bu[2], Kiran A Vaidya[1], Natarajan V Bhanu[3], Benjamin A Garcia[3], Crina M Nimigean[1,4,5], Roderic G Eckenhoff[2], Paul M Riegelhaupt[1]*

[1]Department of Anesthesiology, Weill Cornell Medical College, New York City, United States; [2]Department of Anesthesiology and Critical Care, University of Pennsylvania, Philadelphia, United States; [3]Epigenetics Program, Department of Biochemistry and Biophysics, Perelman School of Medicine, University of Pennsylvania, Philadelphia, United States; [4]Department of Physiology and Biophysics, Weill Cornell Medical College, New York City, United States; [5]Department of Biochemistry, Weill Cornell Medical College, New York City, United States

**Abstract** K2P potassium channels are known to be modulated by volatile anesthetic (VA) drugs and play important roles in clinically relevant effects that accompany general anesthesia. Here, we utilize a photoaffinity analog of the VA isoflurane to identify a VA-binding site in the TREK1 K2P channel. The functional importance of the identified site was validated by mutagenesis and biochemical modification. Molecular dynamics simulations of TREK1 in the presence of VA found multiple neighboring residues on TREK1 TM2, TM3, and TM4 that contribute to anesthetic binding. The identified VA-binding region contains residues that play roles in the mechanisms by which heat, mechanical stretch, and pharmacological modulators alter TREK1 channel activity and overlaps with positions found to modulate TASK K2P channel VA sensitivity. Our findings define molecular contacts that mediate VA binding to TREK1 channels and suggest a mechanistic basis to explain how K2P channels are modulated by VAs.

*For correspondence:
par9082@med.cornell.edu

**Competing interests:** The authors declare that no competing interests exist.

## Introduction

Tandem pore (K2P) potassium channels are a group of physiologically important K+ leak channels that modulate cellular resting membrane potential to control excitability (*Enyedi and Czirják, 2010*). Human K2P channel abnormalities cause a wide range of physiological syndromes, including cardiac conduction abnormalities that lead to ventricular tachycardia and fibrillation (*Gurney and Manoury, 2009*; *Friedrich et al., 2014*; *Decher et al., 2017*), familial pulmonary arterial hypertension (*Ma et al., 2013*), salt-sensitive hyperaldosteronism (*Davies et al., 2008*), and familial migraines (*Lafrenière et al., 2010*; *Royal et al., 2019*), underscoring roles for K2P channels in human cardiac, vascular, renal, and pain physiology. K2Ps are proven contributors to the potency of volatile anesthetics (VAs), with K2P knockout animals exhibiting a measurable resistance to VA-induced loss of consciousness (*Heurteaux et al., 2004*; *Pang et al., 2009*; *Steinberg et al., 2015*). Systemic effects of VA administration have been directly linked to K2P channel activity, including a role for carotid body TASK channels in mediating VA-induced respiratory depression (*Cotten, 2013*; *Chokshi et al., 2015*) and for TREK1 channels in the vasodilatory and neuroprotective effects of both polyunsaturated fatty acids and VAs (*Blondeau et al., 2007*; *Tong et al., 2014*).

Initial evidence for a VA-induced potassium leak current was found in molluscan pacemaker neurons, an activity subsequently determined to be mediated by a member of the TASK K2P subfamily

(*Franks and Lieb, 1988*; *Andres-Enguix et al., 2007*). Follow-up studies of both snail and human TASK channels identified specific amino acids that alter TASK responsiveness to VAs or eliminate stereoselective discrimination between enantiomeric VA agents (*Andres-Enguix et al., 2007*; *Conway and Cotten, 2012*; *Luethy et al., 2017*). These findings enabled predictions of structural determinants of anesthetic binding within the TASK channel family (*Bertaccini et al., 2014*). However, significant differences in the gating and regulation of divergent K2P subfamilies make it difficult to generalize molecular details of the interactions between VAs and TASK K2P channels to other anesthetic-sensitive K2Ps (*Patel et al., 1999*), most notably the TREK1 channel.

TREK1 is a member of a subgroup of K2Ps (including TREK1, TREK2, and TRAAK) regulated by the biophysical properties of their surroundings, including membrane stretch or deformation, temperature, and bilayer lipid composition (*Honoré, 2007*). This pronounced sensitivity of TREK1 to its surrounding environment is unique amongst known biologically relevant VA targets. VAs are small hydrophobic drugs that readily partition into lipid bilayers and in vivo anesthetic potency has long been known to correlate with lipid solubility, a relationship that holds true over a wide range of structurally diverse anesthetic agents (*Sonner and Cantor, 2013*). While the majority of studies exploring the interaction between anesthetics and ion channels have identified discrete binding sites thought to underlie the modulatory effects of these drugs (*Nury et al., 2011*; *Sauguet et al., 2013*; *Hemmings et al., 2019*), modulation of TREK1 by VAs could be posited to occur via indirect effects on the properties of the surrounding bilayer. In fact, a recent study has proposed that in vitro administration of VA agents disrupt membrane lipid rafts, altering phospholipase C activity and modifying the milieu of lipids surrounding TREK1 to modulate its activity (*Pavel et al., 2020*).

The question of how VA drugs modulate TREK1 is the key issue addressed in this study. By utilizing an unbiased photolabeling approach, we show that the VA isoflurane binds to a TREK1 TM2 residue in close proximity to the previously proposed VA modulatory site in TASK K2P channels. Molecular dynamics (MD) simulation studies based on our photolabeling findings show that isoflurane, a known low-affinity ligand for TREK1, is mobile within this newly identified TREK1-binding site and interacts with residues from TM2, TM3, and TM4. These interactions include key molecular contacts with residues known to influence the conformational rearrangement of the TM4 helix that drive K2P modulation by heat, mechanical stretch, and other pharmacological channel modulators (*Dong et al., 2015*; *Lolicato et al., 2014*; *Brohawn et al., 2014a*; *Pope et al., 2018*). Our photolabeling and MD results enabled us to design multiple point mutations within the putative VA-binding pocket that specifically diminished TREK1 VA responsiveness without otherwise perturbing channel function. Our data support the notion of a shared VA modulatory site across the TREK and TASK K2P subfamilies and suggest a mechanism by which binding of VA within this site promotes conformational rearrangements known to gate K2P channels.

## Results

### Azi-isoflurane photolabeling of TREK1 identifies VA-binding sites

In order to identify TREK1 VA-binding sites, we utilized a previously validated photoreactive analog of the clinically important VA isoflurane (*Eckenhoff et al., 2010*). This biochemical adduct, azi-isoflurane, features a diazirine moiety capable of generating a highly reactive and chemically non-selective carbene adduct when irradiated with UV light. Azi-isoflurane has been shown to retain the anesthetic effects of isoflurane in animals, and we first sought to ensure that the chemical modifications present in the azi-isoflurane compound would not alter the effect of this drug on the TREK1 channel. Two electrode voltage clamp studies of *Xenopus laevis* oocytes expressing mouse TREK1 (mTREK1) showed that application of azi-isoflurane causes a dose-dependent potentiation of mTREK1, with an EC$_{50}$ of $735 \pm 192$ μM (*Figure 1*). We found no statistically significant difference between the effect of saturating doses of isoflurane (*Patel et al., 1999*) or azi-isoflurane on mTREK1 channel function (*Figure 1C*).

For photolabeling studies, recombinantly expressed, purified and liposome reconstituted zebrafish TREK1 (drTREK1) protein was used, chosen for its biochemical tractability (*Figure 2—figure supplement 1*). Pairwise alignment of the mouse and zebrafish TREK1 protein primary sequences show 78.3% sequence identity and 89.1% sequence similarity (*Figure 3—figure supplement 1*). The drTREK1 channel has previously been shown to respond to multiple TREK1 modulatory cues

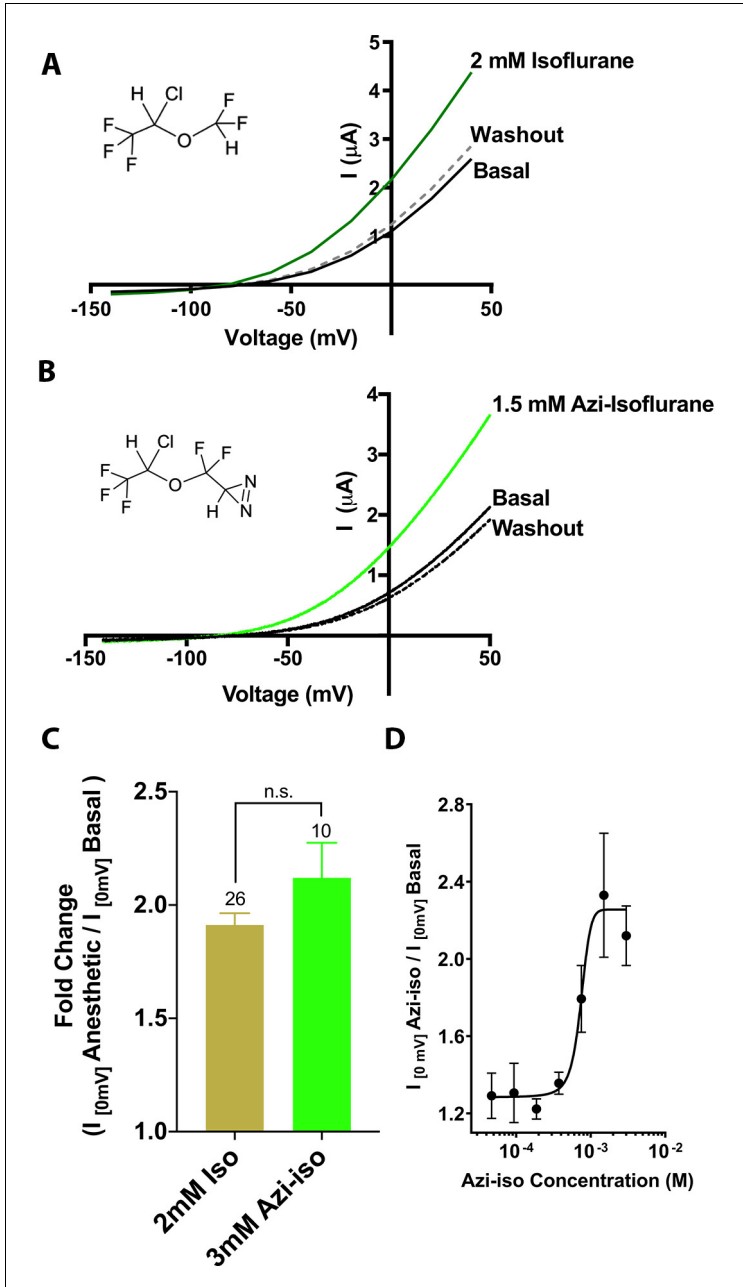

**Figure 1.** Functional validation of azi-isoflurane activity on TREK1 channels. Representative two electrode voltage clamp recordings demonstrate the potentiating effects of saturating doses of isolflurane (**A**) and azi-isoflurane (**B**). Chemical structures of isoflurane and azi-isoflurane are shown. (**C**) Fold effect of administration of saturating doses of either isoflurane or azi-isoflurane on TREK1 outward current, as determined by the ratio of the recorded current at a voltage of 0 mV, immediately prior to and following administration of volatile anesthetic (VA) agent. No significant difference was found between the responses of TREK1 to isoflurane versus azi-isoflurane, unpaired two tailed t-test p value of 0.11 (**D**) Dose response curve for azi-isoflurane activation of TREK1. Data derived from n > 6, N > 2 experimental observations. Error bars in panels C and D are mean ± SEM.

including arachidonic acid and mechanical stretch (*Brohawn et al., 2014b*), and we found no evidence of differences in the functional behavior of the two orthologues with respect to modulation by azi-isoflurane, temperature, or the TREK1 activator BL1249 (Figure 3 and *Figure 3—figure supplement 1*), the modulatory cues explored in our study.

drTREK1 was reacted with 30 μM azi-isoflurane, a concentration well below the EC$_{50}$ of azi-isoflurane for mTREK1, chosen to minimize non-specific modification. Following photolabeling, mass spectrometry (MS) of the drTREK1 protein showed evidence of adduction of azi-isoflurane at two residues, G182 and K194, both located on the second transmembrane domain (TM2) of the channel (*Figure 2*). To examine whether the clinically relevant parent VA isoflurane also binds at these two sites, we performed a parallel azi-isoflurane photolabeling study of drTREK1 in the presence of 3 mM isoflurane as a competitive inhibitor. The 100-fold excess concentration of isoflurane protected the G182 site from azi-isoflurane photolabeling but did not prevent labeling at K194 (*Figure 2—figure supplements 2* and *3*), suggesting that only the G182 site is specifically occupied by the parent VA isoflurane.

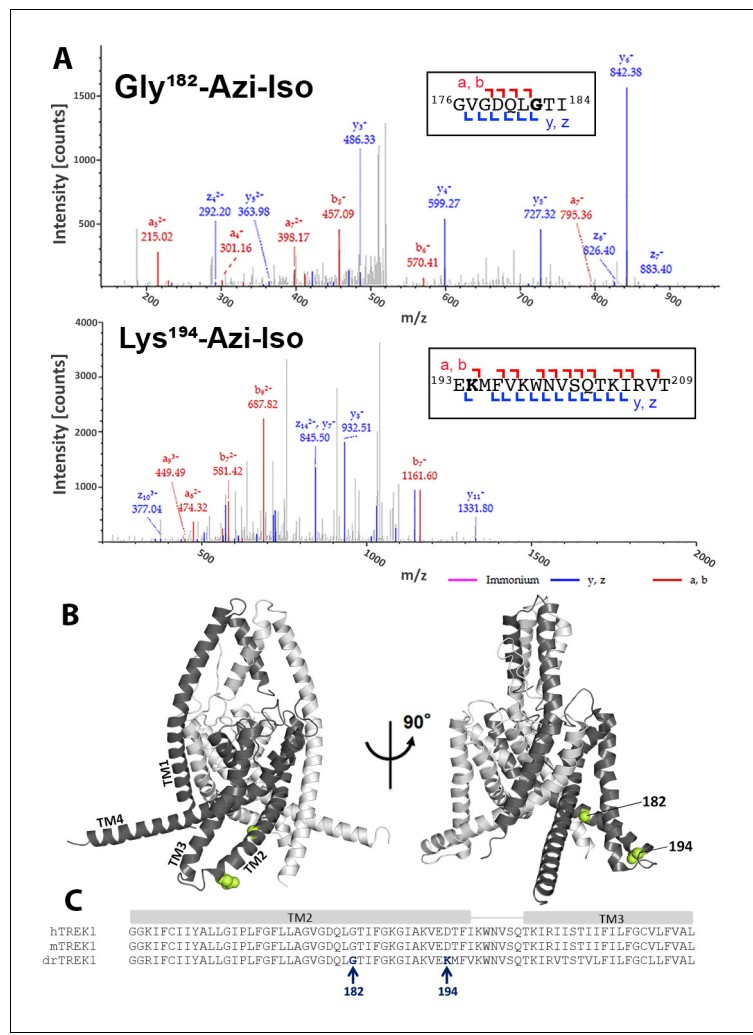

**Figure 2.** Azi-isoflurane photolabeling of TREK1. (**A**) Mass spectra of drTREK1 photoaffinity labeled peptides labeled at Glycine182 (top) and Lysine 194 (bottom). Colored intensities denote the identified peptide a, b, z, and y ion fragments for the sequence assignment, as shown in the inset boxes. See *Figure 2—figure supplement 3* for corresponding peptide tables. (**B**) A structural model of mouse TREK1 (PDBID 6CQ6), showing the positions of residues G182 and K194 (labeled lime spheres) along the TM2 helix. (**C**) Alignment of the TREK1 TM2 and TM3 helixes from human (hTREK1), mouse (mTREK1), and zebrafish (drTREK1).

The online version of this article includes the following figure supplement(s) for figure 2:

**Figure supplement 1.** Purification of TREK1 and TRAAK proteins.

**Figure supplement 2.** Mass spectrometry (MS) analysis of purified drTREK1.

**Figure supplement 3.** Mass spectrometry protein fragment tables.

Although our MS results cover >91% of the drTREK1 protein sequence with high confidence, there are notable regions of the TM3 and TM4 helices that are not identified by MS (*Figure 2—figure supplement 2*). We attributed lack of coverage in these regions to the difficulty in ionizing and resolving hydrophobic transmembrane protein regions with this technique. Alternatively, binding of the hydrophobic azi-isoflurane photolabel could reduce the MS signal from bound protein peptides, causing gaps in MS coverage that correspond to regions where photolabeling has occurred. To exclude this possibility, we performed MS on non-photolabeled drTREK1 protein, finding essentially the same lack of high confidence coverage within the TM3 and TM4 region (*Figure 2—figure supplement 2*). This suggests that the poor MS coverage of drTREK1 TM3 and TM4 is not the result of photolabeling within these regions. However, we cannot rule out the possibility that azi-isoflurane might label positions within the TM3 or TM4 domains that we are simply unable to detect with our MS approach.

## Functional validation of TREK1 VA-binding sites

To determine the functional importance of the VA- binding sites identified by azi-isoflurane photolabeling, we introduced mutations into the mTREK1 gene at positions 182 and 194 and used two-electrode voltage clamp recordings to assay for changes in mTREK1 channel properties. We first substituted the endogenous mTREK1 amino acids at positions 182 and 194 with tryptophan, to mimic the size and hydrophobic nature of the azi-isoflurane photolabel. Tryptophan mutagenesis at position 194 (as well as other more conservative modifications) had no significant effect on the functional properties of the resultant mutant mTREK1 channels (*Figure 3*). The amino acid at position 194 is poorly conserved across species, a lysine in the drTREK1 gene used for azi-isoflurane photolabeling but an aspartic acid in the mTREK1 construct used for our functional studies. To account for this difference, we introduced a D194K mutant into the mTREK1 background and found that this mutation also had no effect on mTREK1 basal current or temperature dependence (*Figure 3E and F*). The absence of any observable functional effect of mutation at mTREK1 194, along with the inability of excess isoflurane to protect position 194 from photolabeling by azi-isoflurane, suggest that despite being modified by azi-isoflurane, this site is unlikely to be relevant to the mechanism by which isoflurane and other clinically relevant VAs modulate TREK1. This notion is supported by the location of position 194 at the far end of TM2, facing toward the bulk solution and away from the core of the TREK1 protein.

TREK1 G182 is located within the center of TM2, one helical turn away from a glycine at position 178 previously shown to be a hinge point for a buckling motion of the TM2 helix that occurs during K2P gating (*Lolicato et al., 2014*). Tryptophan mutagenesis at mTREK1 G182 demonstrated a large increase in basal current density and a near complete loss of modulation by heat, isoflurane, and the TREK1 activator BL1249, findings suggestive of mTREK1 channel activation (*Figure 3A–D*, note that in *Figure 3* panels B-D the concentration of injected cRNA was adjusted to normalize basal current density between mutant and WT mTREK1 channels). Given the unique ability of glycine residues to impart helical flexibility and the known conformational movements in the region of TREK1 around G182 (*Lolicato et al., 2014*; *Brohawn et al., 2014a*; *Dong et al., 2015*), we sought to determine whether loss of flexibility at position 182 was responsible for the major functional effects observed in the mTREK1 G182W mutant. By introducing more conservative mutations at G182, we discovered that the effect of mutagenesis was correlated with the size of the introduced amino acid. mTREK1 G182A showed only a small potentiating effect on current density and no significant effect on gating by heat and mTREK1 G182C showed an intermediate phenotype (*Figure 3E–F*).

To further demonstrate the potentiating effect of increased side chain size at mTREK1 G182, we treated mTREK1 G182C channels with MMTS, a membrane permeant cysteine-modifying reagent (*Figure 4*). In wild-type mTREK1 channels, MMTS treatment caused an increase in channel activity that washed out within 2–3 min of removal of MMTS, a seemingly non-specific effect that occurred even in an mTREK1 construct with all five endogenous cysteines removed (mTREK1 cys-). By contrast, modification of the mTREK1 G182C mutant with MMTS caused potentiation of channel activity that persisted after washout of the MMTS reagent (*Figure 4A*). An equivalent effect was observed in an mTREK1 cys- G182C mutant, specifically confirming that it is modification of the G182 cysteine that leads to the observed persistent potentiation of mTREK1 currents following MMTS application (*Figure 4B*). These findings support the notion that steric crowding is an important determinant of

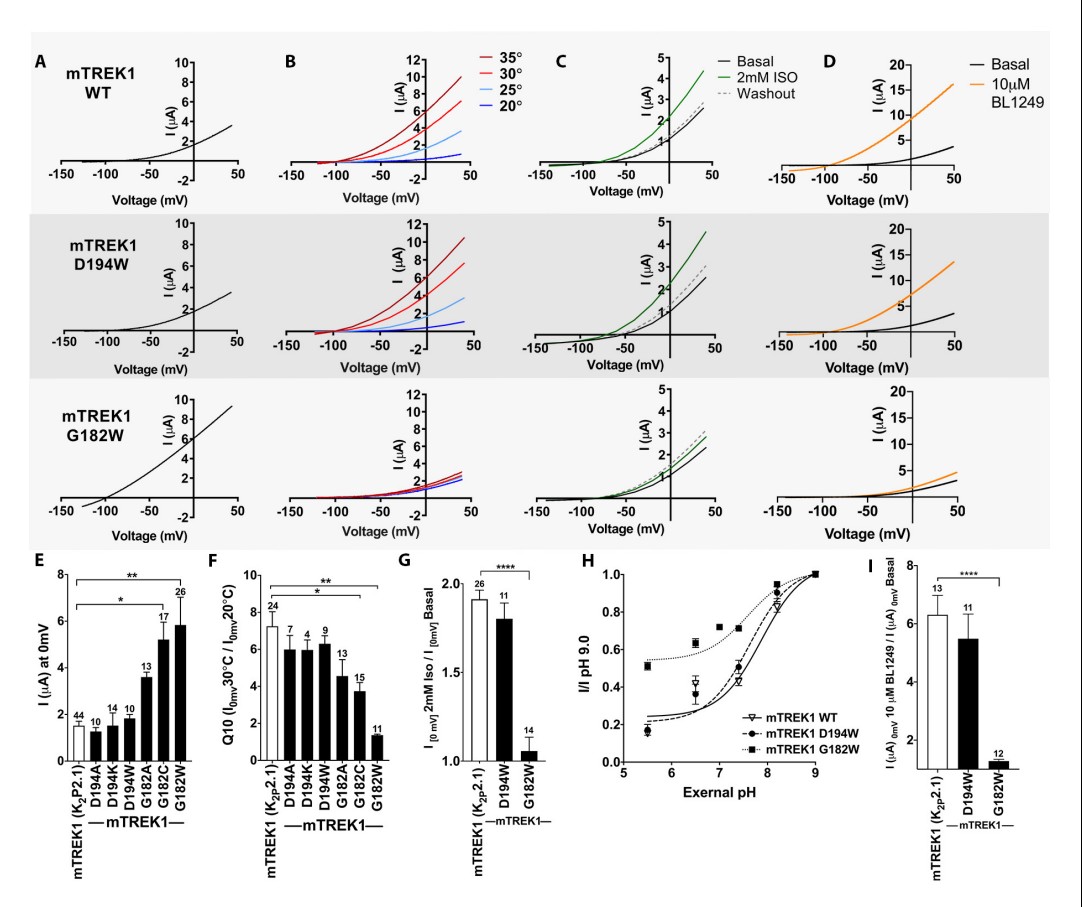

**Figure 3.** Functional assessment of mTREK1 residues identified by azi-isoflurane photolabeling. Representative two electrode voltage clamp recordings of mTREK1 wildtype (WT) and channel mutants G182W and D194W. (**A**) Basal current measured 24 hr after microinjection of 2.4 ng cRNA. (**B**) Temperature dependence of TREK1 currents, measured at temperatures of 20–35°C, in 5°C increments. (**C**) Response to administration of 2 mM isoflurane, followed by washout. (**D**) Response to administration of 10 µM BL1249. For temperature, isoflurane, and BL1249, experiments performed on TREK1 channels bearing mutations that alter basal current density, the concentration of microinjected cRNA was titrated to achieve 1 µA of current at 20°C, to approximate WT channel current density. (**E**) Quantification of TREK1 channel activity on basal current level, (**F**) temperature dependence as measured by Q10 (30°C/20°C), (**G**) response to isoflurane administration, (**H**) changes in external pH, or (**I**) BL1249 administration, as measured by TREK1 current at 0 mV. Number of replicate experiments indicated. Statistical significance was determined by one-way ANOVA combined with a Dunnetts multiple comparison test against mTREK1 WT data, results indicated, *p<0.5, **p<0.05, ****p<0.0005. Error bars are mean ± SEM.

The online version of this article includes the following figure supplement(s) for figure 3:

**Figure supplement 1.** Function assessment of the drTREK1 isoform.

the potentiation observed after perturbation of the G182 residue, whether by mutagenesis, MTS reagent modification, or by occupancy with a VA agent or photolabel.

To explore whether disruption of isoflurane binding contributes to the VA insensitivity of the G182W mutant, we performed azi-isoflurane photolabeling studies on purified drTREK1 G182W protein. Similar to our functional results in the mTREK1 background, the drTREK1 G182W mutant also exhibits diminished sensitivity to heat, VA, and BL1249 (*Figure 3—figure supplement 1* panels F–H). We found no evidence of azi-isoflurane labeling at position 182 or at any neighboring residues (*Figure 2—figure supplement 2*), suggesting that the G182W mutation either eliminates azi-isoflurane binding or reduces the affinity of VA for this site. We did observe azi-isoflurane photolabeling at two positions far from the G182 site, one near the top of TM1 (A67) and one in the C-terminal domain (T303). These residues were not photolabeled in the VA-sensitive drTREK1 WT background and as such they are unlikely to contribute to the mechanism by which VAs modulate TREK1. However, enhanced accessibility of these residues in only the G182W background could be explained by

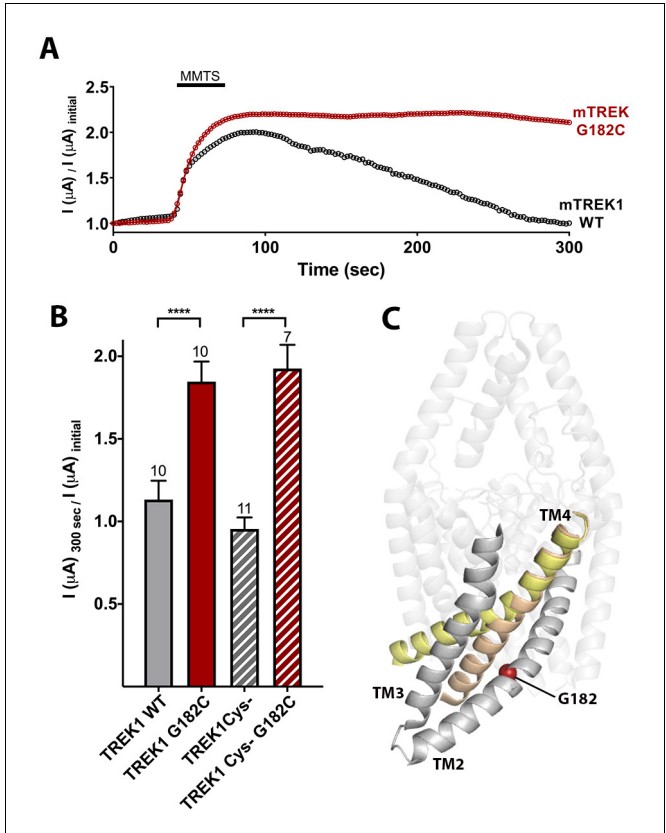

**Figure 4.** Cysteine modification at the G182 position leads to TREK1 activation. (**A**) Representative time courses of the response of TREK1 wildtype (WT) or TREK1 G182C channels to treatment with 2 mM MMTS. Shown is the recorded current at 0 mV holding potential, measured every 5 s for 5 min, with current values normalized to the initial value recorded at the beginning of the time course (**B**) Quantification of effect size at the end of the time course (300 s), for TREK1 WT and TREK1 Cys-, a mutant TREK1 channel lacking all five of the endogenous TREK1 cysteine residues, and for G182 mutants in both of the above backgrounds. Number of replicate experiments indicated. Error bars are mean ± SEM. Statistically significance determined by one-way ANOVA combined with a Sidak multiple comparison test of TREK1 WT versus TREK1 G182C and TREK1 Cys- versus TREK1 cys- G182C. Results indicated, ****p<0.0005. (**C**) Crystallographically defined structural models of TREK2 in the TM4 up (PDB ID 4xdl, yellow) and TM4 down (PDB ID 4bw5, tan) conformations, highlighting the TM2, TM3, and TM4 helices from only a single subunit. The position of G182 (red sphere) is noted.

global changes in protein tertiary structure induced by the G182W mutation, consistent with the dramatic functional effects observed in this mutant (*Figure 3* and *Figure 3—figure supplement 1*).

While the G182 residue is located at the cytosolic face of the TREK1 channel, activation gating in K2P channels is thought to occur via a 'C-type' gating mechanism involving a rearrangement of the extracellular selectivity filter region (*Bagriantsev et al., 2011*; *Piechotta et al., 2011*; *Zilberberg et al., 2001*). Given that the G182 residue is located far from the selectivity filter (*Figure 2B*), a significant allosteric effect would be required for steric crowding at G182 to exert an effect on this distant gate. To explore whether the G182W mutation modulates the TREK1 'C-type' gate, we examined the stability of the selectivity filter against closure by application of extracellular acidosis. Acidification of the extracellular face of TREK1 has been shown to inhibit channel activity and to simultaneously diminish potassium selectivity, a hallmark of selectivity filter-based gating (*Cohen et al., 2008*; *Sandoz et al., 2009*). While the mTREK1 WT and mTREK1 D194W mutants were both strongly inhibited by application of extracellular acidosis, the mTREK1 G182W channel was resistant to closure by acid (*Figure 3H*, $IpH_{5.5}/pH_{9.0}$ 0.15 ± 0.05 for mTREK1 WT, 0.17 ± 0.1 for mTREK1 D194W, 0.51 ± 0.08 for mTREK1 G182W), although the $IC_{50}$ of the effect was similar for all three channels tested (pH 7.88 ± 0.07 for mTREK1 WT, 7.63 ± 0.08 for mTREK1 D194W, and 7.59 ± 0.07 for mTREK1 G182W). In line with these observations, the mTREK1 G182W channel was

also resistant to the effects of BL1249 (*Figure 3D and I*), a TREK1 activator believed to act by promoting potassium ion occupancy within the selectivity filter (*Schewe et al., 2019*). These data support a mechanistic model in which isoflurane modulation of the TREK1 channel occurs via anesthetic binding to the G182 site causing allosteric stabilization of the selectivity filter 'C-type' gate, which leads to increased TREK1 channel activity.

Many of the biophysical modalities known to modulate TREK1 channels, including heat, mechanical stretch, intracellular acidosis, and bioactive lipids, are believed to alter channel activity by affecting the selectivity filter 'C-type' gate (*Bagriantsev et al., 2011*; *Bagriantsev et al., 2012*; *Piechotta et al., 2011*), though the input sensor for these biophysical gating cues is thought to be the intracellular C-terminal domain (*Honoré et al., 2002*; *Chemin et al., 2005b*; *Bagriantsev et al., 2012*). Structurally defined 'TM4 up' and 'TM4 down' conformational states *Brohawn et al., 2014a*; *Lolicato et al., 2014*; *Dong et al., 2015* have been identified as the key rearrangements that allow these C-terminal domain inputs to traverse the protein and converge at the selectivity filter to modulate gating (*Zhuo et al., 2016*). Within this gating model, the location of the G182 anesthetic-binding site along the TM2 helix in direct opposition to TM4 (*Figure 4C*) is intriguing. It suggests that VA occupancy at the G182 site could modulate TREK1 activity by influencing TM4 intramolecular rearrangements known to play a key role in K2P gating. To further explore the isoflurane-binding site and gain insight into the effect of anesthetic binding on TREK1 channel gating, we utilized MD simulation.

## MD simulation identifies residues important for VA modulation to TREK1 channels

In order to establish a suitable starting point for MD simulations, we evaluated potential ligand binding configurations compatible with our photoaffinity results, using Autodock Vina docking software (*Trott and Olson, 2010*). A pocket formed by G182 and the nearby TM3 and TM4 helices had the highest ranked score and was the only predicted binding site near the G182 residue. This binding site is located entirely within the transmembrane region of mTREK1 and isoflurane is a relatively low-affinity ligand with minimal electrostatic interactions. Our docking results confirmed that the binding site near G182 was a sterically favorable starting point to place isoflurane for subsequent equilibrium MD simulation. Simulations of mTREK1 embedded in a POPC:cholesterol lipid bilayer were conducted of mTREK1 wildtype (WT; 440 ns) and mTREK1 G182W in the absence of isoflurane (960 ns), and of mTREK1 WT with a single isoflurane molecule placed at the G182 site in one of the two K2P subunits (two trajectories, 700 ns and 1200 ns). A representative MD snapshot of the isoflurane-binding site is depicted in *Figure 5A*.

Simulation of WT mTREK1 in the presence of VA demonstrated isoflurane to be highly mobile within the binding pocket, adopting a range of positions (*Figure 5B*) and ultimately escaping the binding site after approximately 360 ns of simulation during trajectory 1. This observed high degree of isoflurane mobility is consistent with both the low binding affinity predicted by docking analysis and the high concentrations of isoflurane required to activate TREK1 in functional assays. To prevent escape of the isoflurane molecule from the binding pocket and better sample its bound state during trajectory 2, we imposed a flat-well spherical restraint as used in *Salari et al., 2018* to keep the isoflurane molecule within a sphere of radius 7 Å in the G182-binding pocket. This restraint only imposed an energetic cost in the rare instance when the ligand escaped its site and migrated to the edge of the sphere (*Figure 6—figure supplement 1*, panel A), and zero cost otherwise. The isoflurane ligand center of geometry exceeded 7 Å from the flat-bottomed restraint center to interact with the restraint wall 0.0265% of the time or 1489 out of 5,628,007 snapshots of the restraint coordinate (sampled every 200 fs), showing that the restraint had a minimal energetic effect on the simulation.

The two independent MD trajectories converge on a number of high occupancy positions (*Table 1*, *Figure 5A*), with occupancy defined as the percentage of MD trajectory snapshots in which the isoflurane ligand was within 7 Å of a given residue. These positions include the G182 residue and numerous additional amino acids on mTREK1 TM2, TM3, and TM4. In agreement with our photolabeling findings, MD simulations demonstrated that isoflurane remained in close proximity to the G182 residue for >94% of the time VA occupied the binding site. This finding supports the notion that isoflurane imposes a consistent steric crowding at the G182 position, akin to the activity enhancing modifications we introduced through either mutagenesis or biochemical modification.

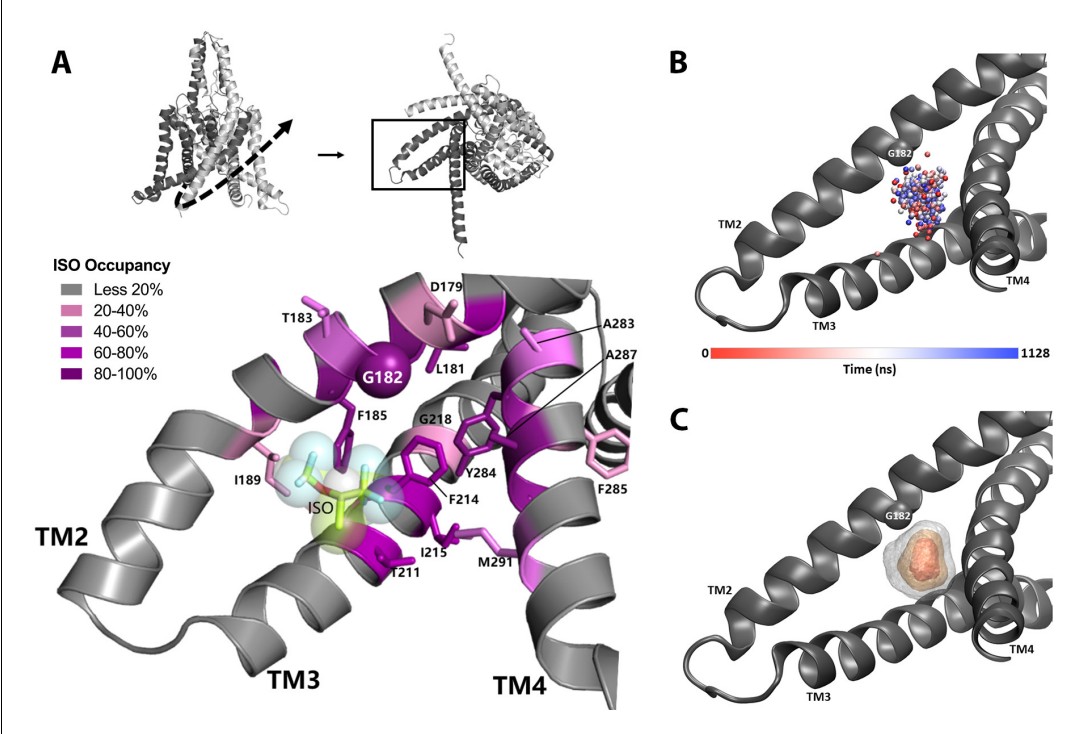

**Figure 5.** Molecular dynamics (MD) simulation studies define an isoflurane-binding pocket in TREK1 channels. (**A**) By rotating the TREK1 structure shown in *Figure 2* (top left, dashed arrow), we demonstrate the region of TM2, TM3, and TM4 that forms the isoflurane-binding site (top right, boxed). Below, a representative MD simulation snapshot of the isoflurane binding pocket, highlighting the G182 residue (sphere) and additional positions found to exhibit high isoflurane occupancy (sticks). Relative isoflurane occupancy for each residue (see *Table 1*) is shown as quartiles, as described on left (**B**) Position of the oxygen atom at the center of the isoflurane molecule during equilibrium MD simulation trajectory 2 of mTREK1 WT in the presence of isoflurane (**C**) Density map of the position of the bound isoflurane. Isosurfaces represent 10% (gray), 30% (yellow), and 50% (orange) isoflurane occupancy.

The proximity of the VA-binding site to the neighboring TM4 helix (*Figure 4C*) suggested a mechanism by which anesthetic binding could influence K2P activity by influencing TM4 position, a concept bolstered by our MD simulation data. We found a number of TM4 residues that directly interact with the bound isoflurane molecule, most notably Y284 and M291. mTREK1 Y284, the only polar residue in the otherwise hydrophobic TM4 helix, has been proposed to form stabilizing hydrogen bonds with one of two alternative TM3 backbone carbonyls in the 'up' versus 'down' conformations of TM4 (*Dong et al., 2015*) but is highly occupied by isoflurane when the drug is bound in the VA site. Similarly, the M291 position has been proposed to form TM4 conformation-dependent interactions with neighboring TM3 and TM4 residues and we found that M291 is occupied by the bound isoflurane molecule during both of the simulation trajectories.

To assess for structural changes in the mTREK1 protein induced by the presence of bound isoflurane, we compared the MD simulations of isoflurane-free versus isoflurane-bound mTREK1 and calculated the per residue root-mean-square deviation (RMSD) of the mTREK1 protein during the two simulations, with respect to the last frame of the WT-free mTREK1 simulation. The isoflurane molecule did not induce any major conformational changes in mTREK1 during either of the simulation trajectories, and we did not observe any alteration in the position of the TM4 helix attributable to the presence of isoflurane. However, we found localized increases in RMSD within the VA-binding pocket at mTREK1 F185 and I189 (TREK1 + Isoflurane MD trajectory 1) or F214 (TREK1 + Isoflurane MD trajectory 2) (*Figure 6a*). These localized deviations in RMSD were absent when we limited our analysis to the main chain Cα carbon (*Figure 6—figure supplement 1B*), and we found that side-chain rotameric orientation was clearly altered at these positions by the presence of the isoflurane ligand (*Figure 6C* and *Figure 6—figure supplement 1D*).

**Table 1.** Occupancy of residues in G182 isoflurane-binding pocket, defined as the percentage of snapshots where isoflurane within 7 Å of the given residue during two independent molecular dynamics (MD) simulation trajectories.

Residues with occupancy of less than 20% in both trajectories are omitted. TREK1 residues homologous to positions previously found to mediate volatile anesthetic (VA) sensitivity in TASK K2P channels are annotated (*).

| Mouse TREK1 residue number | Percent occupancy during MD trajectory 1 | Percent occupancy during MD trajectory 2 | Transmembrane domain |
|---|---|---|---|
| GLY178 | 47.65% | 79.20% | TM2 |
| ASP179 | 35.19% | 35.36% | TM2 |
| LEU181 | 68.40% | 91.72% | TM2 |
| GLY182 | 94.01% | 98.65% | TM2 |
| THR183 | 69.07% | 41.84% | TM2 |
| PHE185* | 92.17% | 97.89% | TM2 |
| GLY186 | 89.16% | 57.30% | TM2 |
| ILE189 | 52.96% | 10.07% | TM2 |
| THR211* | 56.20% | 88.69% | TM3 |
| PHE214 | 44.39% | 82.77% | TM3 |
| ILE215 | 25.65% | 95.46% | TM3 |
| GLY218 | 9.25% | 61.26% | TM3 |
| ALA283 | 28.07% | 61.74% | TM4 |
| TYR284 | 67.95% | 96.27% | TM4 |
| PHE285 | 27.41% | 41.74% | TM4 |
| ALA287 | 88.05% | 99.48% | TM4 |
| VAL288 | 78.20% | 94.79% | TM4 |
| MET291* | 56.11% | 37.02% | TM4 |

The F185 residue has previously been shown to impact K2P responsiveness to heat, VAs, mechanical stretch, and the pharmacological efficacy of BL1249, a K2P activator (*Lolicato et al., 2014*; *Dong et al., 2015*; *Luethy et al., 2017*; *Pope et al., 2018*) and is known to form a pi-stacking interaction with the TM3 F214 residue (*Figure 6B*). This F185/Y214 interaction has been proposed to stabilize the 'TM4-down' gating conformation in TREK2 channels (*Dong et al., 2015*). To explore the effect of isoflurane binding on this structurally important interaction, we quantified pi-stacking between F185 and F214 in our simulations, such that the two aromatic rings were considered to be pi-stacked if their centroids were separated by 4.4 Å or less and the angles between the ring planes were less than 30° (*McGaughey et al., 1998*). In the mTREK1 WT and mTREK1 G182W simulations, pi-stacking between these residues occurred in 37% and 38% of trajectory frames, respectively, but pi-stacking occurred in less than 2% of the trajectory frames in both of the isoflurane-bound wild-type mTREK1 simulations (*Figure 6D*). In TREK1 + isoflurane MD simulation trajectory 1, the isoflurane ligand directly interrupts the pi-stacking interaction (*Figure 6B and F*), while in trajectory two it is the ligand induced reorientation of the F214 rotamer that is responsible for disrupting the pi-stacking interaction (*Figure 6G*). Even after the isoflurane molecule escapes from its binding site during trajectory 1 (*Figure 6D*), the F185/F214 interaction remained disrupted due to the isoflurane induced reorientation of the F185 sidechain.

While additional sampling time might have allowed the F185/F214 interaction to eventually re-establish, the structural model that serves as the starting point for our simulations is in the mTREK1 'TM4-up' state and the F185/F214 interaction is predicted to favor the 'TM4-down' conformation of mTREK1. In the neighboring unliganded subunit of mTREK1, we found no evidence of F185/F214 pi-stacking, lending support to the idea that pi-stacking between these two residues is generally disfavored in the 'TM4-up' state (*Figure 6—figure supplement 1C*). The observed difference in the behaviors of the two subunits likely reflects subtle differences in initial conditions derived from differences in crystal packing in the regions surrounding the TM2/TM3 loops of the two subunits. Given

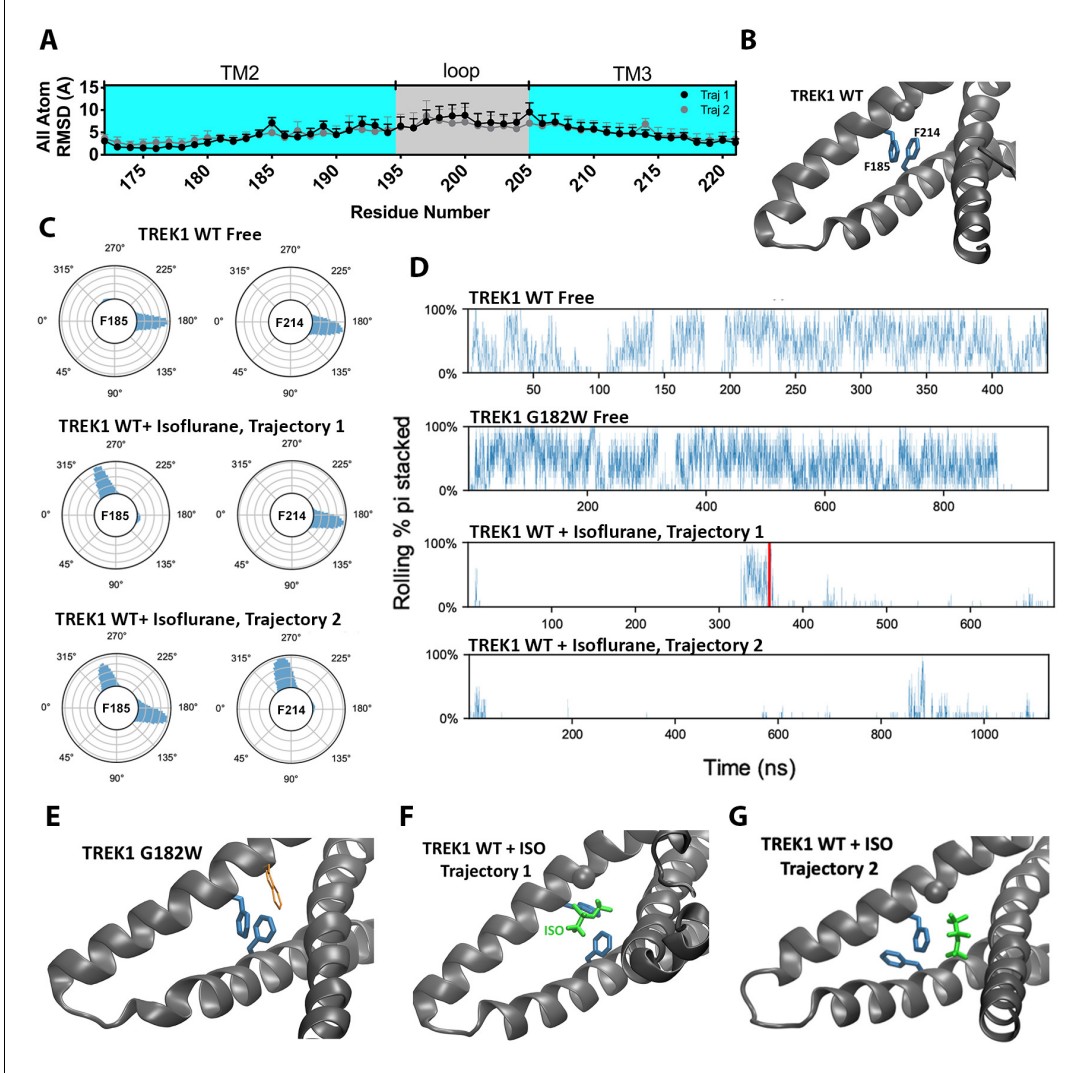

**Figure 6.** The presence of the isoflurane ligand disrupts a key TM2/TM3 interaction. (**A**) All atom per residue RMSD within the TREK1 TM2/TM3 loop from TREK1 wildtype (WT) + Isoflurane Trajectory 1 (black symbols) or Trajectory 2 (gray symbols), as compared with the final frame of the equilibrated TREK1 WT-Free simulation. (**B**) A representative equilibrium MD simulation snapshot of TREK1 WT. The G182 residue is represented as a sphere and the pi-stacking interaction between F185 and F214 is shown. (**C**) Side chain $\chi_1$ dihedral angle residence plots for residue F185 and F214 during the TREK1 WT-free (top) and TREK1 WT + isoflurane simulation trajectories (middle, bottom) (**D**) Pi-stacking plots for the F185/F214 residue interaction, examined by sampling the average number of pi-stacked snapshots over a rolling window of 10 snapshots spanning every 200 fs across the entire simulation timescale. The red bar in the TREK1 WT + Isoflurane trajectory one panel represents the time when isoflurane escaped the binding pocket during this simulation. (**E**) Representative equilibrium MD simulations snapshots of G182W, showing retained pi-stacking of the F185 and F214 residues or (**G,H**) simulation snapshots of TREK1 in the presence of isoflurane with disrupted F185/F214 pi-stacking.

The online version of this article includes the following figure supplement(s) for figure 6:

**Figure supplement 1.** Additional molecular dynamics simulation data.

the relatively short simulation timescales sampled in our study, we treated each subunit independently and did not expect to observe significant allosteric effects across the two subunits.

Based on our MD findings, we hypothesize that a combination of VA-induced local structural perturbations, disruption of the F185/F214 pi-stacking interaction, and direct steric effects upon TM4 movement all combine to alter the dynamics of the TM4 movements known to influence K2P gating. While our MD simulations did not directly reveal major conformational rearrangements of TM4, the observed isoflurane occupancy at numerous positions known to govern the energetics of TM4

translocation from the 'down' to 'up' position suggest a mechanism by which VA binding ultimately alters K2P activity by modulating TM4 movements.

## Key residues transfer VA insensitivity from TRAAK to TREK1

Although both TREK and TASK channels are potentiated by VA agents, the TRAAK K2P channel is anesthetic insensitive (*Patel et al., 1999*). TRAAK is 44% identical and 69% homologous to TREK1, and the two channels share many functional properties, including modulation by heat, mechanical stretch, pH, and arachidonic acid (*Maingret et al., 1999*; *Kang et al., 2005*). The relative selectivity of VA agents for TREK1 over TRAAK despite the many similarities between these two channels allowed us to utilize the TRAAK channel as a negative control to further characterize the K2P VA-binding site.

VA insensitivity of TRAAK could arise from one of two possibilities; failure of VAs to bind to TRAAK or inability of bound VA to modulate TRAAK activity. To distinguish between these possibilities and ascertain whether VAs bind to the TRAAK channel, we recombinantly expressed, purified, and reconstituted human TRAAK (hTRAAK) into liposomes (*Figure 2—figure supplement 1*) and performed azi-isoflurane photolabeling studies identical to those used for drTREK1 channels. The G182 residue identified by azi-isoflurane photolabeling of drTREK1 is conserved in the hTRAAK sequence, but we found no evidence of azi-isoflurane photolabeling at this glycine or at any other position in the hTRAAK channel (*Figure 7—figure supplement 1*). MS of hTRAAK protein after photolabeling showed a relatively lower total coverage of the hTRAAK sequence compared to our results for drTREK1 (84% for hTRAAK versus 91% for drTREK1), but all hTRAAK residues homologous to the drTREK1 azi-isoflurane binding region identified by MD simulations were positively identified, precluding the possibility that azi-isoflurane binds within this region of hTRAAK. Our results suggest that the inability of VA agents to activate TRAAK channels is due to either an absence of binding or a significantly reduced affinity of anesthetic for this VA binding region.

We next explored the sequence similarity between mTREK1 and mTRAAK at the residues that showed the highest isoflurane occupancy during MD simulations. Amongst the identified amino acids, we found four residues that differ significantly between the mTREK1 and mTRAAK sequences (*Figure 7A–B*) and created a series of mutant mTREK1 channels that contain one or two of the corresponding mTRAAK residues at these positions (*Figure 7C–E*). When examined by TEVC, the 'TRAAK-like' mutant mTREK1 channels all exhibit diminished responsiveness to the potentiating effect of isoflurane (*Figure 7C*), supporting a role for these residues in either forming the VA binding domain or transducing the effects of binding to channel opening. While the extent of the effect of these mutants was less dramatic than the near complete absence of isoflurane responsiveness in the G182W mutant, the effect of combining two TRAAK-like mutations was additive, as shown for the double mutant mTREK1 F185L G186R (*Figure 7C*).

Although the G182W mutant exhibited diminished responsiveness to both VA activation and heat, the 'TRAAK-like' mutants maintain their sensitivity to activation by heat (*Figure 7D–E*). The fact that the 'TRAAK-like' mutants selectively alter VA activation without significantly perturbing gating by heat argues for their specificity to the mechanism of action of VAs. Activation of TREK1 by heat is known to be critically related to the phosphorylation state of serine 333 in the intracellular C-terminal domain of TREK1 (*Maingret et al., 2000*) and the persistence of mTREK1 heat activation in these 'TRAAK-like' mutants suggests that these mutations do not eliminate VA modulation by simply causing a global activation of the mTREK1 'C-type' gate (as was the case for the G182W mutant). Rather, the 'TRAAK-like' mutations appear to reduce VA sensitivity by acting specifically at the VA site of action, consistent with predictions made by our MD simulation data and mTREK1 vs mTRAAK sequence homology.

## Discussion

Effort to determine regions within the K2P channel structure susceptible to small-molecule modulators has become a topic of significant recent interest (*Dong et al., 2015*; *Lolicato et al., 2017*; *Pope et al., 2018*; *Rinné et al., 2019*), motivated by the lack of specific and high-affinity pharmacology targeting K2P channels. Here, we focus on K2P modulation by VAs, drugs heavily utilized in common clinical practice. We identify an interface formed by the TM2, TM3, and TM4 helices of TREK1 as the binding site for the VA isoflurane and identify site-specific interactions between TREK1

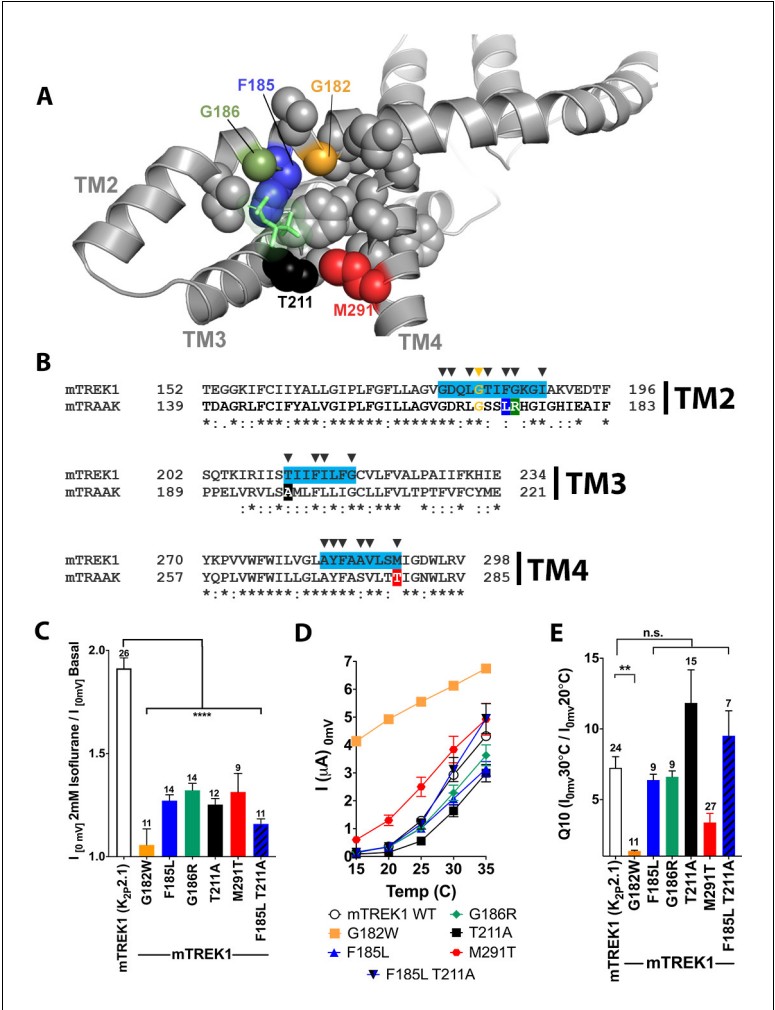

**Figure 7.** Mutations in the isoflurane-binding site alter anesthetic sensitivity without perturbing global channel function. (**A**) Representative MD) snapshot showing the isoflurane-binding site, including residues predicted to have >20% occupancy by isoflurane shown in sphere representation and isoflurane shown in stick representation (lime). (**B**) Alignments of mouse TREK1 and TRAAK sequences, with the isoflurane-binding domain regions of TREK1 TM2, TM3, and TM4 (as identified by MD simulation) highlighted in blue. Arrows denote positions of high isoflurane occupancy. Poorly conserved residues are color coded throughout the figure [F185 (blue), G186 (green), T211 (red), and M291 (black)]. (**C**) Quantification of TREK1 wildtype and mutant responses to 2 mM isoflurane administration, (**D**) temperature, as measured by TREK1 current at 0 mV, or (**E**) temperature dependence as measured by Q10 (30°C/20°C). Number of replicate experiments indicated. Error bars are mean ± SEM. Statistically significance was determined by one-way ANOVA combined with a Dunnetts multiple comparison test against mTREK1 WT data, results indicated, **p<0.05,****p<0.0005.

The online version of this article includes the following figure supplement(s) for figure 7:

**Figure supplement 1.** Mass spectrometry (MS) analysis of purified human TRAAK.

and isoflurane that define this binding region. The identified VA modulatory site features TREK1 residues previously identified to play key roles in K2P modulation by mechanical stretch, heat, and the pharmacological activator BL1249, suggesting a common mechanistic pathway shared by VA and other K2P gating cues.

Prior to our study, the strongest evidence for a VA-binding site in K2P channels was found in the TASK subfamily, where a single point mutation in the *lymneal* TASK TM3 residue M159 (homologous to TREK1 position T211) caused loss of steroselective discrimination between R and S optical isomers of isoflurane (*Andres-Enguix et al., 2007*). TREK1 T211 lies within a span of the TM3 helix we were unable to resolve with MS but through MD simulation we identified T211 as a determinant of

TREK1 VA binding (*Table 1*). Two additional residues identified by MD simulation, TREK1 F185 and M291, also align to positions implicated in TASK VA sensitivity (*Andres-Enguix et al., 2007*; *Conway and Cotten, 2012*). All three residues (F185, T211, M291) differ significantly in the TRAAK protein sequence and 'TRAAK-like' mutations at these positions decreased TREK1 VA sensitivity (*Figure 7B*). The overlapping molecular determinants of TREK1 and TASK1 VA sensitivity (and of TRAAK VA insensitivity) suggest a shared VA-binding site across the K2P family. While the gating mechanisms of TASK channels have not been as extensively studied as those of the mechanosensitive K2Ps, recent work exploring the inhibitory effect of the local anesthetic agent bupivacaine on TASK1 channels demonstrates that bupivacaine binding constrains TM4 helix movement and ultimately alters selectivity filter behavior (*Rinné et al., 2019*). The bupivacaine site of action in TASK channels differs from the VA modulatory site identified in our study, but the effect of drug binding on TM4 conformation may be a shared feature in K2P gating utilized by both local and general anesthetics.

Our data suggest that VAs modulate K2P channel activity by influencing the position of the TM4 helix. The proximity of the azi-isoflurane labeled TREK1 G182 residue to the TM4 helix initially suggested such a mechanism and the potentiating effect of increased G182 side chain size supported this hypothesis. MD simulation studies identified direct contacts between isoflurane and the TM4 helix. However, the MD simulation data do not show major TM4 conformational rearrangements in the presence of isoflurane and thus whether VAs induced TM4 movements. Similarly, we have no evidence of whether the VA-bound TREK1 channel more closely resembles the 'TM4 up' or 'TM4 down' states defined by crystallography (*Dong et al., 2015*; *Brohawn et al., 2014a*). As there is evidence to suggest that both the 'TM4 up' and 'TM4 down' conformations can promote active K2P channel states (*McClenaghan et al., 2016*), each of the two conformations could be consistent with a VA-activated channel. However, both the direct steric constraints imposed by the presence of VA near TM4 and the disruption of the 'TM4 down' state stabilizing F185/F214 pi-stacking interaction would be predicted to favor a 'TM4 up' conformation. The starting point for our MD simulations is a crystallographically determined structural model of TREK1 already in the 'TM4 up' state (*Lolicato et al., 2017*) and may explain the lack of any additional TM4 rearrangement in the presence of isoflurane. Further MD simulation utilizing a homology model of TREK1 based on K2P structures determined in the 'TM4 down' state might more definitively answer this question. However, it is also possible that VA binding induces a conformation distinct from either of these two structurally defined states. The molecular mechanisms governing the connectivity between TM4 conformation and the flux gating behavior of the selectivity filter (*Schewe et al., 2016*) remains poorly defined and is an area of significant ongoing interest.

Having identified a VA modulatory site that appears to be shared across the TREK and TASK K2P subfamilies, we are cognizant of some of the inconsistent VA responses observed across these K2P subfamilies. For instance, while the obsolete VAs chloroform and diethyl-ether activate TREK1 and TASK3, they paradoxically inhibit the TASK1 channel (*Patel et al., 1999*; *Andres-Enguix et al., 2007*). Nitrous oxide and xenon have been reported to activate TREK1 but have no effect on TASK channels (*Gruss et al., 2004*), and TRAAK channels are unresponsive to VAs despite their close similarity to other mechanosensitive and VA-sensitive K2Ps. Meanwhile, little is known about the gating of THIK channels but they are inhibited by the VAs halothane and isoflurane, an effect that appears to play an important role in the clinically relevant central respiratory depressant effect of VA administration (*Rajan et al., 2001*; *Lazarenko et al., 2010*). How can we explain the paradoxical effects of VA agents at structurally similar targets? One explanation would be that the VA-binding site identified in our study represents only one of a subset of VA modulatory domains in the K2P family. Within the anesthetic mechanisms literature there are numerous examples of VA-responsive ion channels that feature multiple low-affinity anesthetic modulatory sites acting in a combinatorial fashion to produce a given output (*Fourati et al., 2018*; *Heusser et al., 2018*; *Woll et al., 2018*; *Hemmings et al., 2019*). However, the location of the identified K2P VA-binding site at a position central to TM4 gating offers an alternative explanation. We understand relatively little about the mechanism by which TM4 movements govern K2P channel activity and it is possible that VA binding to this single site may produce opposing effects in the context of subtle differences in the connectivity between TM4 movements and the channel gate. Recent studies utilizing the state-dependent binding of the mechanosensitive K2P inhibitor fluoxetine demonstrate that the resting and activated conformation of the TM4 helix in TREK1 versus TRAAK channels are inverted, underscoring this point

(*Soussia et al., 2018*) and perhaps contributing to the absence of VA binding and responsiveness in TRAAK channels.

The TREK1 channel is strongly modulated by its surrounding membrane environment, with lipid headgroup as well as acyl-chain length and saturation playing roles in shaping K2P functional behavior (*Chemin et al., 2005a*; *Chemin et al., 2005b*; *Cabanos et al., 2017*). In prokaryotic ligand gated ion channels, binding sites for anesthetics and lipids have been found to overlap (*Nury et al., 2011*; *Prevost et al., 2012*), raising the question of whether the VA-binding site identified in our study could also be a site responsible for lipid modulation of TREK1. While numerous structural studies of K2P channels have identified bound lipids, none of these lipid-binding sites overlap with the VA-binding site identified in our study (*Brohawn et al., 2014a*; *Dong et al., 2015*; *Lolicato et al., 2017*). Despite this absence of structural evidence, an MD simulation study of TREK2 has shown that a lipid can bind to a site akin to our TREK1 VA-binding site and that this bound lipid restricts the movement of TM4 in response to a simulated membrane stretch (*Aryal et al., 2017*). While we suspect that multiple sites on the TREK1 channel may ultimately play important roles in the modulatory effects of lipids, numerous residues within the VA-binding site are likely to be determinants of both lipid and VA modulation of K2P channel gating. The interplay between the mechanisms by which VAs and lipids modulate the TREK1 channel is an area of ongoing interest for future study.

While our results support the presence of a defined modulatory site within the TREK1 channel structure, these findings do not preclude a contribution from indirect effects of VAs on membrane lipid composition, as has been recently proposed (*Pavel et al., 2020*). Such a mechanism is in fact a particularly appealing explanation for the effects of the structurally simple and TREK1-specific VA agents chloroform, nitrous oxide, and xenon (*Gruss et al., 2004*). In this context, the location of the TREK1 VA-binding site described in our study at a region key to the mechanism by which C-terminal-domain-dependent signals (including lipid modulation) are translated to TM4 (*Zhuo et al., 2016*) and ultimately to the selectivity filter gate would allow both an indirect and direct modulatory effect on channel behavior to feed into a shared common pathway.

## Materials and methods

### Purification of drTREK1 or hTRAAK

TREK1 and TRAAK proteins were expressed in *Pichia pastoris* using previously described pPICZ vectors bearing residues 1–300 of the human TRAAK gene (hTRAAK, K2P4.1, UniProt Q9NYG8-2) or residues 1–322 of the *D. rerio* TREK1 gene (drTREK1, K2P2.1, UniProt X1WC65), followed by a PreScission protease-cleavage site (LEVLFQ/GP) and C-terminal GFP and His$_{10}$ tags (*Brohawn et al., 2014b*). Mutations to eliminate N-linked glycosylation sites were inserted into both the hTRAAK (N104Q and N108Q) and drTREK1 (N95Q, N122Q) gene constructs. Expression plasmids were linearized with the PmeI restriction enzyme and were subsequently transformed by electroporation into *P. pastoris* strain SMD1168H. Screening for successful recombinant integration was performed by plating transformants on yeast extract peptone dextrose sorbitol (YPDS) plates containing increasing concentrations of zeocin from 0.5 to 3 mg/mL.

Screening, expression, and purification of K2Ps was performed as previously described (*Lolicato et al., 2014*). Briefly, yeast transformants were grown in buffered minimal medium (2 × YNB, 1% glycerol, 0.4 mg L$^{-1}$ biotin, 100 mM potassium phosphate [pH 6.0]) for 2 days at 30°C in a shaker at 225 rpm. Cells were then pelleted by centrifugation (4000 × g at 20°C, 5 min) and resuspended in methanol minimal medium (2 × YNB, 0.5% methanol, 0.4 mg L$^{-1}$ biotin, 100 mM potassium phosphate [pH 6.0]) to induce protein expression. Cells were then shaken for 2 additional days at 22°C in a shaker at 225 rpm, with additional methanol added (final concentration 0.5% [v/v]) to the culture after 24 hr of protein expression. After 48 hr of protein expression, cells were pelleted by centrifugation (6000 × g at 4°C for 10 min), flash frozen in liquid nitrogen, and then subjected to three rounds of cryo-milling (Retsch model MM301) in liquid N$_2$ for 3 min at 25 Hz to disrupt yeast cell walls and membranes. Frozen yeast cell powder was then stored at −80°C.

Cell powder was added to breaking buffer (150 mM KCl, 50 mM Tris pH 8.0, 1 mM phenylmethylsulfonyl fluoride, 0.1 mg/mL DNase 1, and one tablet/50 mL of EDTA-free complete inhibitor cocktail [Roche]) at a ratio of 1 g cell pellet/4 mL lysis buffer. Solubilized cell powder was centrifuged at 4000 × g at 4°C for 5 min to pellet large debris and the supernatant was then centrifuged at

100,000 × g at 4°C for 1.5 hr to pellet cell membranes. The pellet was re-suspended in 50 mL breaking buffer containing 60 mM n-Dodecyl-B-D-Maltoside (DDM) and incubated for 3 hr with gentle stirring to solubilize the membranes, followed by centrifugation at 35,000 × g for 45 min. Talon cobalt resin (Takara Bio USA) was added to the supernatant at a ratio of 1 mL of resin per 10 g of cell powder and incubated in an orbital rotor overnight at 4°C. Resin was then collected on a column and washed with 10 column volumes of Buffer A (150 mM KCl, 50 mM Tris pH 8.0, 6 mM DDM, 30 mM imidazole) and bound protein was subsequently eluted from the resin by washing with Buffer B (150 mM KCl, 50 mM Tris pH 8.0, 6 mM DDM, 300 mM imidazole). PreScission protease (~1:25 wt:wt) was added to the eluate and the cleavage reaction was allowed to proceed overnight at 4°C under gentle rocking. Cleaved drTREK1 or hTRAAK protein was concentrated in 50 kDa molecular weight cutoff (MWCO) Amicon Ultra Centrifugal Filters (Millipore) and applied to a Superdex200 10/300 gel filtration column (GE Healthcare) equilibrated in size exclusion chromatography (SEC) buffer (150 mM KCl, 20 mM Tris pH 8.0, 1 mM DDM). Purified hTRAAK or drTREK1 protein was concentrated (50 kDa MWCO) to 8 mg/mL prior to reconstitution and analyzed for purity by SDS-PAGE [12% (wt/vol) gels; Bio-Rad] followed by staining with coomassie blue. All protein purification steps were carried out at 4°C.

## Reconstitution of K2P channels into liposomes

Immediately following purification, drTREK1 or hTRAAK channel protein was reconstituted into liposomes (*Heginbotham et al., 1998*). Five mg of a 3:1 (wt/wt) ratio of 1-palmitoyl-2-oleoyl-sn-glycero-3-phosphoethanolamine (POPE) and 1-palmitoyl-2-oleoyl-sn-glycero-3-phospho- (1'-rac-glycerol) (POPG) in chloroform was used for reconstitution. The lipids were dried in a borosilicate glass vial under nitrogen flow and then solubilized in reconstitution buffer (400 mM KCl, 10 mM HEPES, 5 mM NMDG [N-methylglucamine D-gluconate], pH 7.6) containing 34 mM CHAPS (3-((3-cholamidopropyl) dimethylammonio)−1-propanesulfonate). The lipid containing solution was bath sonicated until clear. drTREK1 or hTRAAK protein was then added to the lipid solution at a concentration of 100 μg protein/mg of solubilized lipids. Proteoliposomes were formed by applying the lipid/protein containing sample (500 μL total volume) to an 18 mL detergent-removal column (Sephadex G-50 fine beads, GE Healthcare Life Sciences). Turbid fractions containing proteoliposomes were pooled, aliquoted, flash frozen in liquid nitrogen, and stored at −80°C.

## Photoaffinity labeling of drTREK1 or hTRAAK potassium channels for protein microsequencing

A total of 5–7 μg of drTREK or hTRAAK in proteoliposomes was added to a 30 μM solution of azi-isoflurane ±3 mM isoflurane. Each sample was equilibrated on ice in the dark for 5 min and then transferred to a 1 mm path length quartz cuvette and exposed for 25 min to 300 nm ultraviolet light produced by an RPR-3000 Rayonet lamp filtered by a WG295 295 nm glass filter (Newport Corporation).

## In-solution protein digestion

Photolabeled samples underwent dialysis and buffer exchange using 10 kDa MWCO Amicon Ultra Centrifugal Filters (Millipore). ProteaseMAX Surfactant (Promega) was added to a concentration of 0.2%, and the samples were vortexed vigorously for 30 s. Samples were then diluted with $NH_4HCO_3$ to a final concentration of 50 mM $NH_4HCO_3$ in a 93.5 μL volume. 1 μL of 0.5 M dithiothreitol (DTT) was then added and samples were incubated at 56°C for 30 min. 2.7 μL of 0.55 M iodoacetamide (IAA) was subsequently added and protein samples were incubated at room temperature in the dark for 45 min. An additional 1 μL of 1% (w/v%) ProteaseMax Surfactant was added to the sample, followed by sequencing grade-modified trypsin (Promega) to a 1:20 protease:protein final ratio (w:w). Proteins were digested overnight at 37°C. Trypsin digested peptides were again diluted with $NH_4HCO_3$ to a final concentration of 100 mM $NH_4HCO_3$ and 0.02% ProteaseMAX Surfactant in a 200 μL total volume, prior to addition of sequencing grade chymotrypsin (Promega) to a final 1:20 protease:protein ratio (w:w). Proteins were again digested overnight at 37°C. Acetic acid was added to reach a pH <2 and the peptide digests were then incubated at room temperature for 10 min, prior to centrifugation at 16,000 x g for 20 min to remove insoluble debris. Samples were then

desalted using C18 stage tips prepared in house, dried by speed-vac, and resuspended in 0.1% formic acid immediately prior to MS analysis.

## In-gel protein digestion

Photolabeled samples underwent dialysis and buffer exchange using 10 kDa MWCO Amicon Ultra Centrifugal Filters (Millipore). Samples were then mixed with SDS loading buffer containing DTT to a final concentration of 100 mM DTT, vortexed vigorously and incubated at room temperature for 45 min before separation by SDS-PAGE. The resulting gels were stained with Coomassie Blue G250 (BioRad), destained, and washed with ddH$_2$O. Protein bands between ~30 and 40 kDa (corresponding to *dr*TREK or *h*TRAAK) were identified and excised. Excised bands were destained, dehydrated and dried by speed vac. Proteins were then reduced by incubation at 56˚ C for 30 min in 5 mM DTT and 50 mM NH$_4$HCO$_3$. The DTT solution was removed and proteins were then alkylated by the addition of 55 mM IAA in 50 mM NH$_4$HCO$_3$ and incubation at room temperature for 45 min in the dark. Bands were dehydrated and dried by speed vac before resuspension in 100 µL 0.2% ProteaseMAX surfactant and 50 mM NH$_4$HCO$_3$ solution containing trypsin at a 1:20 protease:protein ratio (w:w). Proteins were digested overnight at 37˚C. To enhance sequence coverage, a second protease digestion was performed with chymotrypsin. Samples were diluted to a final volume of 200 µL with final concentrations of 100 mM NH$_4$HCO$_3$ and approximately 0.02% ProteaseMAX Surfactant. Sequencing grade chymotrypsin (Promega) to a 1:20 protease:protein ratio (w:w) was then added and proteins were digested overnight at 37˚C. Multiple extraction steps were performed to remove peptides embedded in the gel. The digest solution was removed and the remaining gel was suspended in 100 µL 30% acetylnitrile and 5% acetic acid in ddH$_2$O (v/v%) and sonicated for 20 min. The solution was removed and the gel was then resuspended in 100 µL 70% acetylnitrile and 5% acetic acid in ddH$_2$O (v/v%) and sonicated for 20 min. The digest and two extraction solutions were pooled and dried by speed vac before resuspension in 0.5% acetic acid (pH <2). Samples were sonicated for 10 min prior to centrifugation to remove insoluble debris. Samples were desalted using C18 stage tips prepared in house. Samples were dried by speed vac and resuspended in 0.1% formic acid immediately before MS analysis.

## Mass spectrometry

Desalted peptides were analyzed employing either an Orbitrap Elite Hybrid Ion Trap-Orbitrap mass spectrometer (MS) or Q Exactive Hybrid Quadrupole-Orbitrap MS coupled to an Easy-nanoLC 1000 system. In both instances, the same liquid chromatography procedure and data-dependent acquisition mode was applied. Peptides were eluted over 100 min with linear gradients of ACN in 0.1% formic acid in water (v/v%) starting from 2% to 40% (85 min), then 40% to 85% (5 min) and finally 85% (10 min) using a flow rate of 300 mL/min. For Orbitrap Elite, in every 3 s cycle, one full MS scan was collected at a scan range of 350–1500 m/z, a resolution of 60K, and a maximum injection time of 50 ms with automatic gain (AG) control of 500,000. The MS2 scans were followed from the most intense parent ions. Ions were filtered with charge 2–5 with an isolation window of 1.5 m/z in quadruple isolation mode. Ions were fragmented using collision induced dissociation (CID) with collision energy of 35%. Iontrap detection was used with normal scan range mode and rapid iontrap scan rate. For Q Exactive, one full MS was performed with 70K resolution, maximum injection time of 100 ms and scan range of 350–1200 m/z. The MS2 scans were performed in Higher-energy Collisional Dissociation (HCD) with normalized collision energy (NCE) of 30, and isolation window of 3 m/z. The AG control was set to be 10,000 with a maximal injection time of 100 ms.

## Mass spectrometry analysis

Analysis was performed as previously reported (*Eckenhoff et al., 2010*; *Woll et al., 2018*). Spectral analysis was conducted using Thermo Proteome Discoverer 2.0 (Thermo Scientific) and the Mascot Daemon search engine with a customized database containing *dr*TREK or *h*TRAAK protein sequences. All analyses included dynamic oxidation of methionine (+15.9949 *m/z*) as well as static alkylation of cysteine (+57.0215 *m/z*; iodoacetamide alkylation). Photolabeled samples were run with the additional dynamic azi-isoflurane (+195.97143 *m/z*) modification. A mass variation tolerance of 10 ppm and 20 ppm for MS and 0.8 and 0.02 Da for MS/MS were used for the for Orbitrap Elite and Q Exactive, respectively. Both the in-solution and in-gel sequential trypsin/chymotrypsin digests were

searched without enzyme specification with a false discovery rate of 0.01. All MS experiments were conducted in triplicate and samples containing no photoaffinity ligand were processed and analyzed equivalently to those containing the photolabel, to control for false-positive detection of photoaffinity ligand modifications.

## Molecular dynamics simulation

The crystallographically derived structural model of *Mus musculus* TREK1 (PDBID 6CQ6) was used for MD simulation. Missing nonterminal loops were predicted using MODELLER (*Webb and Sali, 2017*) with side-chain conformations predicted using SCRWL4 (*Krivov et al., 2009*). Molecular mechanics parameters for isoflurane were used, as previously published (*Hénin et al., 2010*). In the ligand-bound simulation, one isoflurane molecule was placed manually, adjacent to G182, in the position of the best-scoring hit from molecular docking of isoflurane to TREK1 using AutoDock Vina software (*Trott and Olson, 2010*). Docking parameters were: num_modes = 9, exhaustiveness = 8, energy_range = 3. The search box was 43.5 × 40.0 × 30.0 Å in size, centered on one of the two G182 residues. Docking scores of the nine hits requested from AutoDock Vina ranged from −5.1 to −4.6. Three of these were adjacent to G182, including the best-scoring hit.

The mTREK1 channel was embedded in a POPC (1-palmitoyl-2-oleoyl-*sn*-glycero-3-phosphocholine):cholesterol 70:30 lipid bilayer and the system was solvated with TIP3P water with 0.15 M NaCl using the CHARMM-GUI service (*Jo et al., 2008*; *Jo et al., 2009*). The CHARMM36 force field (*Klauda et al., 2010*; *Best et al., 2012*) and NAMD 2.12 simulation software (*Phillips et al., 2005*) were used. Periodic boundary conditions with particle mesh Ewald summation of long-range electrostatics was used. The system was equilibrated according to the CHARMM-GUI protocol and production equilibrium MD simulation was run for 10 ns using the isothermic-isobaric ensemble with Langevin thermostat and barostat at 303.15 K. Further equilibrium MD simulations were continued from that point.

During isoflurane-bound mTREK1 WT simulation trajectory 2, a flat-well spherical restraint as used in *Salari et al., 2018* was imposed to keep the isoflurane molecule within a sphere of radius 7 Å in the G182 binding pocket while minimizing the energy imparted in the system to do so. This restraint only imposed an energetic cost if the ligand migrated to the edge of the sphere, and zero cost otherwise. As this restricts only the volume the ligand can occupy, it represents purely an entropic cost which can be quantified parametrically as $\Delta G_{restraint}$ = -$RT$ ln $V^*/V$ where $V^*$ is the volume occupied by the sphere, $V$ is the volume occupied by one ligand molecule at 1 M concentration, $R$ the gas constant and $T$ the temperature. For the 7 Å radius spherical flat-well restraint, $\Delta G_{restraint}$ = 0.09 kcal/mol.

For root-mean-square deviation (RMSD) calculations to assess binding site changes induced by isoflurane, both of the isoflurane-bound mTREK1 WT simulation trajectories were analyzed using the last frame of the WT-free simulation as reference. To remove global rotation and translation, simulation trajectories were aligned at selectivity filter residues I143, F145, and N147.

## Two electrode voltage clamp electrophysiology

For electrophysiological studies, *Xenopus laevis* oocytes were microinjected with capped RNA translated from full length mouse or zebrafish TREK1 (K2P2.1, UniProt P97438-2) or mouse TRAAK (K2P4.1, UniProt O88454) genes. TREK1 mutations were introduced using a Phusion site-directed mutagenesis kit (ThermoFisher), with the mTREK1 C93S C159S C219S C365S C399S mutant channel referred to as mTREK1 cys- for brevity. cRNA from WT or mutant K2P genes was synthesized using an mMessage mMachine Kit (T7 promoter, Ambion, Life Technologies) and purified using an RNeasy RNA cleanup kit (Qiagen). Defolliculated stage V–VI *Xenopus laevis* oocytes were purchased commercially from Xenoocyte (Dexter, MI) and were stored in antibiotic supplemented ND96 (96 mM NaCl, 2 mM KCl, 1.8 mM CaCl$_2$, 2 mM MgCl$_2$, 100 units mL$^{-1}$ penicillin, 100 μg mL$^{-1}$ streptomycin) until use. Oocytes were microinjected with 2.5 ng cRNA (unless otherwise noted) and two-electrode voltage clamp (TEVC) recordings were performed 24–48 hr after microinjection. All TEVC data in the manuscript represents recordings from oocytes isolated from at least two independent ovaries, n > 3 oocytes per group.

For recordings, oocytes were impaled with borosilicate recording microelectrodes (0.3–2.0 MΩ resistance) backfilled with 3 M KCl. Oocytes were perfused with ND96 solution (96 mM NaCl, 2 mM

KCl, 1.8 mM CaCl2, and 2.0 mM MgCl2, 5 mM HEPES [pH 7.4]) at a rate of 2.5 mL/min. Currents were evoked from a −90 mV holding potential by a 2000 ms ramp from −150 mV to +50 mV. Data were acquired using an OC-725C oocyte clamp amplifier (Warner Instruments) controlled by pClamp software (Molecular Devices), and digitized at 2 kHz using a Digidata 1332A digitizer (MDS Analytical Technologies).

For temperature experiments, recording solutions were heated by an SC-20 in-line heater/cooler combined with an LCS-1 liquid cooling system operated by the CL-100 bipolar temperature controller (Warner Instruments). Temperature was monitored using a CL-100-controlled thermistor placed in the bath solution immediately downstream of the oocyte. The perfusate was warmed from 15°C to 35°C in 5°C increments, with recordings performed once temperature readings stabilized at the desired values. For external pH experiments ($pH_o$), each oocyte was initially perfused with ND96 solution at pH 7.4, switched to ND96 at pH 9.0 and then subsequently assayed at the experimental pH, with recordings performed 15 s after each new solution was applied. This allowed for all $pH_o$-dependent changes to be normalized against the current density at pH 9.0. The buffer present in ND96 solutions was altered as appropriate, with one of the following: 10 mM TRIS (pH 9.0–8.1), 5 mM HEPES (pH 8.1–6.5), or 5 mM MES (pH 6.5–5.9). Aggregate $pH_o$ data were fitted with a modified Hill equation: $I = I_{min} + (I_{max} - I_{min})/(1 + 10((pH_oIC_{50}/pH_o) * H))$, where $I_{max}$ and $I_{min}$ are maximal and minimal current values, respectively, $pH_oIC50$ is a half-maximal effective $pH_o$ value, and H is the Hill coefficient.

For VA experiments, isoflurane saturated ND96 was prepared by adding commercially available isoflurane (Baxter Healthcare Corporation) to a 100 mL volume of ND96 until phase separation was clearly observable. This mixture was stirred vigorously overnight in a closed glass bottle prior to use. The established maximal solubility of isoflurane in salt solution at room temperature is 15 mM (*Scheller et al., 1997*) and working isoflurane concentrations used for electrophysiological experiments were prepared by dilutions of the 15 mM saturated stock solution. Azi-isoflurane was diluted into DMSO prior to use and the concentration of this stock solution was determined using the optical extinction coefficient of 126 $M^{-1}$ $cm^{-1}$ at 300 nm Abs. Anesthetic solutions were stored in closed perfusion bags and were perfused onto oocytes through Teflon perfusion tubing, to prevent loss of isoflurane to the environment or electrophysiology rig.

## Acknowledgements

This work was supported by research grants from the Foundation for Anesthesia Education and Research (FAER) and the NIH National Institute for General Medical Sciences (NIGMS K08 GM132781 and P01 GM055876). We thank Dr. Roderick MacKinnon for generously sharing the yeast expression plasmids necessary for production of drTREK1 and hTRAAK protein and Dr. Philipp Schmidpeter for critically reading the manuscript.

## Additional information

### Funding

| Funder | Grant reference number | Author |
| --- | --- | --- |
| Foundation for Anesthesia Education and Research | FAER-182483-2 | Paul M Riegelhaupt |
| National Institute of General Medical Sciences | K08GM132781 | Paul M Riegelhaupt |
| National Institute of General Medical Sciences | P01GM055876 | Roderic G Eckenhoff |

The funders had no role in study design, data collection and interpretation, or the decision to submit the work for publication.

### Author contributions

Aboubacar Wague, Weiming Bu, Kiran A Vaidya, Natarajan V Bhanu, Investigation; Thomas T Joseph, Kellie A Woll, Formal analysis, Investigation; Benjamin A Garcia, Supervision; Crina M

Nimigean, Conceptualization, Writing - review and editing; Roderic G Eckenhoff, Conceptualization, Supervision, Writing - review and editing; Paul M Riegelhaupt, Conceptualization, Supervision, Funding acquisition, Investigation, Methodology, Writing - original draft, Project administration, Writing - review and editing

### Author ORCIDs
Crina M Nimigean ⓘ http://orcid.org/0000-0002-6254-4447
Paul M Riegelhaupt ⓘ https://orcid.org/0000-0001-8593-2605

### Decision letter and Author response
Decision letter https://doi.org/10.7554/eLife.59839.sa1
Author response https://doi.org/10.7554/eLife.59839.sa2

## Additional files

### Supplementary files
• Transparent reporting form

### Data availability
All data generated or analysed during this study are included in the manuscript and supporting files.

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
