## [Decision Letter]

**Acceptance summary:**

K2P ion channels are fundamental modulators of the resting membrane potential of most cells. These manuscript by Wague et al. presents exciting data that solidly identifies a binding region for volatile anesthetics (VAs) that regulates the function if TREK1 channels. The data paper is supported by various lines of evidence that convincingly present a case for direct gating effects of the volatile anesthetic isoflurane. This is an important finding given that the general mode of action of VAs is still not completely understood. The evidence presented here should help further the case for the existence of direct molecular targets of VAs that modulate cellular excitability and further our knowledge both at the basic and clinical levels.

**Decision letter after peer review:**

Thank you for submitting your article "Mechanistic insights into volatile anesthetic modulation of K2P channels" for consideration by *eLife*. Your article has been reviewed by three peer reviewers, including Leon D Islas as the Reviewing Editor and Reviewer #1, and the evaluation has been overseen by Richard Aldrich as the Senior Editor. The following individual involved in review of your submission has agreed to reveal their identity: Rebecca Howard (Reviewer #3).

The reviewers have discussed the reviews with one another and the Reviewing Editor has drafted this decision to help you prepare a revised submission.

Summary:

This manuscript by Wague et al. presents solid evidence of binding and modulation of TREK1 channels by the volatile anesthetic isoflurane. TREK1 is a member of a family of two-pore potassium channels associated with the resting leak current in most cells, therefore supporting an essential physiological function.

The authors have used photo-affinity labeling of a photo-activated derivative of isoflurane coupled with mass spectroscopy to identify a residue involved in isoflurane binding. Further molecular dynamics and electrophysiology experiments support the finding of a binding pocket in the M2 transmembrane helix and identification of several other residues involved in forming this pocket. The experiments and simulations are supportive of the conclusions reached and the paper is well written. Characterization of a binding site for a volatile anesthetic in leak channels is an important finding that further solidifies the view that volatile anesthetics exert their anesthetic effects by acting of specific ion channels at dedicated binding sites and also helps clarify our understanding of the several anesthetic effects observed in patients and at the cell physiological level.

Essential revisions:

1) Please clarify how the important initial pose for isoflurane was chosen. The Results seem to cite AutoDock Vina, but the Materials and methods indicate the molecule was merely "placed manually, adjacent to G182". Moreover, the Results seem to use docking to substantiate relevance of the G182 site; this claim would benefit from a clear description of the docking approach/parameters, and of other hits and scores obtained.

2) The main limitation of the isoflurane simulations, on which much of the dynamic mechanism relies, is that they represent a single trajectory (albeit of reasonable length) for a single isoflurane molecule (subsection “MD simulation identifies residues important for VA modulation to TREK1 channels”). At least one replicate simulation would be preferable; particularly given isoflurane seems relatively unstable in its docked pose. Why did the authors choose not to dock both subunits?

Similarly, please clarify if the unbound subunit retains its initial configuration (e.g. with regards to π-stacking) in the presence of isoflurane in the neighboring subunit?

3) It is stated that G182W, but none of the "TRAAK-like" mutants, "exhibited diminished responsiveness to…heat" (Figure 6D). To support this comparison, please include statistics indicating a shift in heat sensitivity for G182W but not for other mutants.

4) Given suggestions elsewhere of indirect, lipid-mediated as well as direct mechanisms of anesthetic modulation in these channels, please comment on whether the proposed isoflurane binding site might also be occupied by lipids, in the presence or absence of Vas.

5) Photo labeling and functional studies are carried out in two different orthologues of TREK1. Given that it seems like both are modulated by isoflurane and perhaps equally amenable to the same experimental procedures, please explain this choice.

6) In Figure 1, the potassium current amplitudes at two different concentrations of isoflurane and azi- isoflurane are compared and the authors reached the conclusion that "We found no statistically significant difference between the effect of 2 mM isoflurane or 3 mM azi-isoflurane…". Comparing current size obtained with different concentrations of different compounds is not really significant. What needs to be compared is the complete dose-response relationship of isoflurane and azi- isoflurane. The authors must have these experiments already, as they show the dose-response curve of azi- isoflurane.

7) Since different orthologues of TREK1 are used throughout the manuscript, it would improve the clarity to add the prefix corresponding to the organism in each instance, e.g. dTREK1 for Danio, etc. This is done in some cases in some figures but it should be maintained thorough the manuscript.

8) The data in Figure 3D should not be presented as current magnitude at 0 mV but rather, please use current density, since absolute current size is determined by expression levels as a result of different mutants being trafficked to the membrane differently or changes in open probability and single channel conductance possibly induced by the mutants. At least a control experiment for expression level of the mutants illustrated should be carried out and presented. For example, the current present at 0 mV can be compared to the current activated by a full activator of the channel.

Revisions expected in follow-up work:

Some functional studies in Danio TREK1 should be carried out and shown or cited if available, to demonstrate that at least the same effect of isoflurane and selected mutations (i.e. G182W) is present in this species TREK1 channel. See point 5.

---

## [Author Response]

Essential revisions:1) Please clarify how the important initial pose for isoflurane was chosen. The Results seem to cite AutoDock Vina, but the Materials and methods indicate the molecule was merely "placed manually, adjacent to G182". Moreover, the Results seem to use docking to substantiate relevance of the G182 site; this claim would benefit from a clear description of the docking approach/parameters, and of other hits and scores obtained.

Thank you for alerting us to the need for clarification of this point. A thorough description of the AutoDock Vina approach has now been added to the revised manuscript (subsection “MD simulation identifies residues important for VA modulation to TREK1 channels”). As AutoDock Vina does not produce simulation-ready output on its own, the initial simulation conditions were created by placing an isoflurane molecule in the location identified by the top hit.

We used docking to generate a plausible starting point for dynamics – not, by itself, to substantiate the site. Given the low experimentally-determined affinity of isoflurane for TREK1, we found it unlikely that the initial pose within the experimentally identified site would strongly influence the ensemble of ligand configurations derived from MD simulation.

2) The main limitation of the isoflurane simulations, on which much of the dynamic mechanism relies, is that they represent a single trajectory (albeit of reasonable length) for a single isoflurane molecule (subsection “MD simulation identifies residues important for VA modulation to TREK1 channels”). At least one replicate simulation would be preferable; particularly given isoflurane seems relatively unstable in its docked pose. Why did the authors choose not to dock both subunits?Similarly, please clarify if the unbound subunit retains its initial configuration (e.g. with regards to π-stacking) in the presence of isoflurane in the neighboring subunit?

To address this critique, we have conducted an additional >1μs simulation of TREK1 with isoflurane in the G182 site. To ensure that the isoflurane molecule would not escape from the low affinity TREK1 binding site during this additional trajectory, we introduced a boundary restraint to the ligand. We have included a plot showing the distance of the center-of-geometry of the isoflurane molecule from the center of the restraint sphere (included in new Figure 6—figure supplement 1A), demonstrating that the ligand rarely approached the edge of the restraint during the simulation. A description of this additional restraint is provided in the Materials and methods section of the revised manuscript.

A comparison of residues that exhibit >20% isoflurane occupancy during either the initial MD trajectory or the new 1 μs trajectory is shown in revised Table 1. In general, there is very good concordance between the two simulations and we found no significant differences in the C-α RMSD plots between the two trajectories (shown in Figure 6—figure supplement 1B). In the revised manuscript, we have added an analysis of side chain RMSD, showing significant reorientations of the F185 (trajectory 1) or F214 (trajectory 2) rotameric orientation induced by the presence of the isoflurane ligand. New plots of all-atom RMSD as well as rotamer dihedral angle and pi-stacking frequency for the F185 and F214 residues have all been added to an additional figure in the revised manuscript (new Figure 6).

Given the simulation timescales we were evaluating, we did not expect to observe significant allosteric effects of ligand binding on the neighboring subunit and as such focused our efforts on isoflurane binding to a single subunit, chosen arbitrarily. While we have tried to infer effects of ligand binding on TREK1 channel gating by contextualizing our results within the larger K2P literature, the scope of our simulations are not geared to directly address this question.

In the neighboring subunit, there was virtually no pi-stacking between F185 and F214, in both wild-type free TREK1 simulations and wild-type isoflurane-bound TREK1 simulations. We note that the F185/F214 pi-stacking interaction is predicted to be disfavored in the TREK1 TM4 up conformation present in the only available TREK1 crystal structure. Differences in crystal packing in the regions surrounding the TM2/TM3 loop of the two subunits likely accounts for subtle differences in initial conditions that influence the propensity of these two residues to interact. In the revised manuscript, we note this absence of F185/F214 pi-stacking in the neighboring subunit in the context of a discussion of the failure for the F185/F214 pi-stacking interaction to reform after isoflurane escapes the binding pocket during isoflurane bound TREK1 simulation trajectory 1. We thank the reviewers to directing our attention to this question, one we had not previously explored.

3) It is stated that G182W, but none of the "TRAAK-like" mutants, "exhibited diminished responsiveness to…heat" (Figure 6D). To support this comparison, please include statistics indicating a shift in heat sensitivity for G182W but not for other mutants.

Thank you for pointing out this omission. We have now included Q10 measurements (30°C/20°C) for the "TRAAK-like" mutants as Figure 7E, demonstrating a statistically significant difference in Q10 for TREK1 G182W compared to TREK1 WT and the “TRAAK-like” mutants.

4) Given suggestions elsewhere of indirect, lipid-mediated as well as direct mechanisms of anesthetic modulation in these channels, please comment on whether the proposed isoflurane binding site might also be occupied by lipids, in the presence or absence of Vas.

Thank you for this suggestion. We have now expanded the pertinent section near the end of the Discussion to include this question.

5) Photo labeling and functional studies are carried out in two different orthologues of TREK1. Given that it seems like both are modulated by isoflurane and perhaps equally amenable to the same experimental procedures, please explain this choice.

We agree with the reviewers that performing all of the experiments on a single TREK1 orthologue would have been an ideal experimental approach. While it is clear in retrospect that this would have been possible, at the time it was not clear that the mouse TREK1 construct was amenable to biochemical purification. The decision to use two different orthologues of TREK1 in this study was made entirely for practical considerations, given our knowledge at the outset of the project

The zebrafish TREK1 orthologue utilized for photolabeling studies was chosen for its biochemical tractability. The protocol for expression and purification of the drTREK1 orthologue was published by the MacKinnon laboratory and they graciously shared this construct with us. This drTREK1 protein has previously been characterized (Brohawn PNAS 2014) and shown to be functionally active and modulated by biologically important cues including mechanical stretch and the polyunsaturated fatty acid arachidonic acid.

After we initiated our photolabeling studies, the structure of the mouse TREK1 protein was published by the Minor lab (Lolicato et al., 2017). To date, this is the only published TREK1 structure and as such the mouse TREK1 structural model (PDBID 6cq6) was used as the basis for our molecular dynamics simulations. Functional studies were then performed in the mouse TREK1 background.

In the revised manuscript, we discuss this issue early in the Results section and we have added functional characterization of the drTREK1 orthologue expressed in *X. laevis* oocytes, shown in new Figure 3—figure supplement 1. Similar to the mouse TREK1 orthologue, drTREK1 is modulated by azi-isoflurane, temperature, and BL1249. The drTREK1 G182W mutation blunts responsiveness to all of these modulatory cues.

6) In Figure 1, the potassium current amplitudes at two different concentrations of isoflurane and azi- isoflurane are compared and the authors reached the conclusion that "We found no statistically significant difference between the effect of 2 mM isoflurane or 3 mM azi-isoflurane…". Comparing current size obtained with different concentrations of different compounds is not really significant. What needs to be compared is the complete dose-response relationship of isoflurane and azi- isoflurane. The authors must have these experiments already, as they show the dose-response curve of azi- isoflurane.

We have now amended the manuscript to clarify that in Figure 1, we are comparing the effects of saturating doses of the two drugs. The dose response curve for azi-isoflurane is shown in Figure 1 and we now explicitly cite Patel et al., 1999 as the source for dose response measurement of isoflurane activation of TREK1.

7) Since different orthologues of TREK1 are used throughout the manuscript, it would improve the clarity to add the prefix corresponding to the organism in each instance, e.g. dTREK1 for Danio, etc. This is done in some cases in some figures but it should be maintained throughout the manuscript.

drTREK1 or mTREK1 prefixes have now been added throughout the manuscript, as appropriate.

8) The data in Figure 3D should not be presented as current magnitude at 0 mV but rather, please use current density, since absolute current size is determined by expression levels as a result of different mutants being trafficked to the membrane differently or changes in open probability and single channel conductance possibly induced by the mutants. At least a control experiment for expression level of the mutants illustrated should be carried out and presented. For example, the current present at 0 mV can be compared to the current activated by a full activator of the channel.

We certainly agree that in the ensemble whole cell measurements performed in our study, it is difficult to differentiate between a mutation that alters channel expression level or membrane trafficking from one that alters more fundamental channel properties.

In our initial manuscript, panel 3D was a statistical summary of the results in panel 3A. We injected fixed concentrations of cRNA (as stated in the figure legend) and screened for mutations that alter the measured current density. For the G182W mutant, we found a large increase in outwardly rectifying current density compared to TREK1 WT channels. In order to perform a statistical analysis from recordings of many oocytes, we compared the current recorded at a 0mV holding potential as an arbitrary point far from the reversal potential. This convention has been used frequently by others who study K2P channels, hopefully making it simpler for colleagues to compare their results to our findings.

For the remainder of the experiments in Figure 3, we titrated the amount of G182 cRNA injected into the oocytes to obtain currents with magnitudes similar to TREK1 WT. If the functional consequence of the G182 mutations were solely a result of changes in trafficking or expression, titrating the injected cRNA should account for this and produce recordings indistinguishable from TREK1 WT. On the contrary, we found that the TREK1 G182W mutant showed a near complete loss of positive and negative modulation by heat and a loss of potentiation by the VA isoflurane. In line with the suggestion made by the reviewers, we took these results as a sign that G182W alters intrinsic properties of the TREK1 channel.

In the revised manuscript, we now also include the additional finding that TREK1 G182W is insensitive to saturating doses of BL1249, a potent TREK1 channel activator (Figure 3D and I). We believe the sum of our findings in Figure 3 are most easily explained by an increased open probability in the G182W mutant. We recognize that without single channel recordings we are unable to comment on the effect of the G182W mutant on TREK1 single channel conductance, though there is no evidence in the literature for modulation of TREK1 activity by changes in single channel conductance and as such this seems a less plausible explanation for our observed findings.

Revisions expected in follow-up work:Some functional studies in Danio TREK1 should be carried out and shown or cited if available, to demonstrate that at least the same effect of isoflurane and selected mutations (i.e. G182W) is present in this species TREK1 channel. See point 5.

Please see above, addressed in point 5. This new data is present in Figure 3—figure supplement 1.